# Mineral Layer Fillers for the Production of Functional Materials

**DOI:** 10.3390/ma14123369

**Published:** 2021-06-18

**Authors:** Lidia G. Gerasimova, Marina V. Maslova, Ekaterina S. Shchukina

**Affiliations:** Tananaev Institute of Chemistry of Kola Science Centre, Russian Academy of Sciences, 26a Fersman Street, 184209 Apatity, Russia; l.gerasimova@ksc.ru (L.G.G.); m.maslova@ksc.ru (M.V.M.)

**Keywords:** phlogopite mineral, technogenic waste, electrochemical mica decomposition, electrodialysis of titanium sulfate, hydroxide coating on flakes, shell composition, photocatalyst, pearlescent pigment

## Abstract

An original method based on the use of technogenic waste from the processing of mineral-layered materials, in particular phlogopite for obtaining highly efficient functional compositions of the “mica-TiO_2_”, has been developed. The composition core is a nanosized mica flake coated with mesoporous titanium dioxide of an anatase or rutile structure. Energy-saving and environmentally friendly technological methods are based on the splitting of the mica followed by heterogeneous electrohydrolysis of a mixture of titanium (IV) sulfate solution and flake particles. No destruction of the mica surface, which provided the obtained uniform coatings, has been observed. Such coatings are used in photocatalysis processes and possess a self-cleaning capability. Core–shell compositions are more economically attractive compared with titanium dioxide, in particular TiO_2_ grade P25 (Degusse). The core of the transparent flake and the shell of the rutile titanium dioxide endows the final product with a pearlescent optical effect. This type of material is widely used in the manufacturing of paints and varnishes, printing inks, cosmetics, etc. The use of technogenic waste could significantly reduce the cost of the final product, which would ensure its widespread use in various industries.

## 1. Introduction

Among the known minerals with a layered structure, mainly muscovite (KAl_2_[AlSi_3_O_10_](OH)_2_) and phlogopite (KMg_3_(AlSi_3_O_10_)(OH)_2_) are used, as well as biotite K(Mg,Fe^2+^)_3_(Si_3_Al)O_10_(OH,F)_2_ [1] to a much lesser extent. Phlogopite is the most magnesian mineral of the biotite group. The crystals are tabular, short and prismatic in shape. The crystal structure is a layered lattice. The class of symmetry is prismatic-2/m. Syngony is monoclinic. The spot group is 2/m-prismatic. The spatial group is B2/m (B1 1 2/m) (C2/m) (C1 2/m 1). Muscovite is a rock-forming mineral from the mica subclass of laminated silicates. Muscovite splits easily into thin sheets, due to its crystalline structure, which is composed of three-layer packages of two sheets of silica and alumina tetrahedrons connected through a layer composed of octahedrons, with Al ions in the center, surrounded by four oxygen ions and two OH groups; one third of the octahedrons are not filled with Al ions. The bundles are interconnected with potassium ions. Mica is a raw material for the production of high-temperature electrical insulation for the nuclear and military industries, as well as aircraft construction, instrument making and mechanical engineering. From mica, products are made such as mica paper, mica plastics, mica tapes, mica electric heating elements, and other products for the electrical industry [2,3,4,5,6]. Finely ground mica is added to the coatings of welding electrodes [7]. Specially prepared mica particles are used as carriers for the functional active additives. In particular, powders with a scaly structure of particles are carriers of an oxide functional coating, due to which a pearlescent (interference) effect is formed or photocatalytic properties are acquired. Due to their highly decorative and technical properties, pearlescent powders (PP) are widely used in the paint and varnish industry, as well as in the production of plastics, paper, and in the manufacturing of cosmetics and jewelry [8,9,10]. The big advantage of such PPs in comparison, for example, with the known product consisting of a basic lead carbonate—3PbCO_3_∙2Pb(OH)_2_ [11]—is their non-toxicity, thermal, light and chemical resistance, and low density of 2.9 to 3 g/cm_3_. The size of the lamellar PP particles varies in the range of 5–110 μm, with a coating thickness of 200–600 nm. The most commonly used PP is a composite system “mica-TiO_2_”, in which the core is mica microparticles coated with an oxide compound of titanium (IV). Not only natural materials are used as carriers, but also artificial plates, the synthesis of which is rather complicated. The coating is usually carried out by the method of heterogeneous thermohydrolysis of titanium (IV) sulfate or chloride solution containing mica particles [12,13].

There are several methods for the synthesis of pearlescent pigments, such as the sol-gel method, chemical vapor deposition and the hydrolysis of titanium hydroxide [14,15]. TiO_2_ is known to have the properties of a polymorphic compound and can be represented as rutile, anatase, or brucite, depending on the conditions of its production. They all share the same fundamental structural octahedral units, with a different arrangement of [13]. Rutile is the most stable phase. The refractive index of rutile (2.93) is higher than that of anatase (2.49). Therefore, mica coated with a continuous layer of rutile has a strong luster. Anatase has better photocatalytic properties. Depending on the requirements for pearlescent pigments, a particular titanium dioxide modification is chosen to coat the mica particles. There are known photocatalytic properties of titanium dioxide, in particular TiO_2_ grade P25, manufactured by Degusse [16]. Shell compositions, in which the core is an inert carrier of a mesaporous oxotitanic coating consisting of a TiO_2_ anatase structure, are now being investigated. Mica flakes are also used as carriers. Such compositions are more economical and, when used in products such as building materials, perform the functions of a photocatalyst and promote self-cleaning of the surface [17].

In the specialist literature, not enough attention is paid to the characterization of raw materials for PP production, or to the methods of synthesis of the shell compositions and their influence on the properties of the final product. Furthermore, there are practically no publications that discuss the results of studies on the use of mica production waste.

The purpose of this research is to study and develop a new method for converting the technogenic waste consisting of a phlogopite ore concentration into a flake precursor, and understand its use as a carrier of titanium nanocoating.

## 2. Characteristics of the Raw Material Object and Its Availability

Despite the fairly numerous deposits that have been explored, phlogopite is seldom mined in Russia, but is imported, in particular, from Madagascar. The Kola Peninsula is one of the territories with significant reserves of phlogopite ore. For several decades, phlogopite concentrate was obtained from the Kovdor concentration plant (Kovdor, Murmansk region). Powerful ore-bearing quarries are shown in Figure 1 and Figure 2. At present, the factory has been preserved for a number of objective reasons. The tailings storage of the factory consists of a technogenic deposit with a high content of useful minerals. The “Phlogopite” company, which has now begun to enrich this waste by obtaining a high-quality thermal insulator, has assumed that the amount of raw materials accumulated in the dumps will ensure the operation of the enterprise for several decades. At the same time, reclamation of the technogenic deposit territories will also take place. The idea of reanimating the production of phlogopite and, on its basis, manufacturing marketable products is quite real, since there is a market, as well as appropriate consumers and technologies [5]. However, in addition to resolving the issues related to environmental safety, the issues of import substitutions and the employment of the population will need to be resolved.

From this point of view, it is of interest to use the mica waste not only to obtain the main product—a thermal insulator—but also for obtaining other types of functional materials that are in demand in both the Russian and foreign markets.

## 3. Methods

Mica is an aluminosilicate mineral. The main structural element of mica is a three-layer package of two tetrahedral silicon layers, between which there is an octahedral layer of aluminum and magnesium cations (Figure 3b). Two of the six atoms of oxygen octahedrals are replaced by hydroxyl OH groups. The packets are linked into a continuous structure through K^+^ (or Na^+^) ions, with a coordination number of 12. The layered structure of the mica (Figure 3a) and the weak bond between the packets affects the following properties: lamination, cleavage, strength, elasticity, as well as the ability to split into thin flakes [18].

Since, for the synthesis of the functional materials, the authors proposed using mica waste that had been exposed to the environment for a long time, it is of interest to study the degradation of the mica particles at different pH values.

The object of research was “fresh” waste of phlogopite, taken after the enrichment of mica from the Kovdor deposit. The initial material, consisting of lamellar particles of 10–12 mm, was ground in a porcelain ball mill using the wet mode. A fraction of 63–100 μm was used for the work. The chemical composition of the mica sample was determined using an ICP-QMS mass spectrometer (Perkin Elmer, Waltham, MA, USA) (Table 1), while an image of the mica particles was taken using a Philips XL 30 scanning electron microscope (FEI Company, Netherlands), NS the specific surface area was measured using a Micrometric 2000 instrument (Micromeritics instrument corporation, Norcross, GA USA) by the BET method, based on the “adsorption–desorption of N_2_”. 

The experiment was carried out as follows: A weighed portion of the mica was placed into a closed glass vessel with the solution, in which the pH factor was adjusted within the range of 3–9 using fixanals 0.1 N HCl or 0.05 N NaOH. The resulting suspension with a concentration of 5g/L mica was stirred for 24 h and was then filtered through a membrane filter with a density of 45 μm. The liquid phase was analyzed using the Perkin-Elmer 3100 atomic adsorption spectrometer (Perkin Elmer, Waltham, MA, USA). All the working solutions were prepared using double-distilled water. The initial leaching rate of elements under the selected conditions was calculated by the following formula:$$R=\frac{C\cdot V}{n\cdot m\cdot a\cdot t}$$
where R is the leaching rate of the analyzed element for the selected period of time, mol/(m^2^·s); C is the concentration of the analyzed element in the filtrate, mol/L; V is the volume of the filtrate, L; n is the number of cations of this element in the formula unit; m is the mass of the mineral (mica), g; a is specific surface area of the mineral particles, m^2^/g; and t is the leaching time, days [19].

Technological studies were carried out according to the scheme shown in Figure 4. The method for processing the phlogopite waste included the following operations: splitting the mica, grinding and hydroclassification of the crushed material into fractions, and applying the hydroxide coating to the mica flakes.

An electrochemical method was used to break down the mica particles. The choice of optimal conditions for the experiment is described in detail in [20,21]. Distilled water was used as a reagent. A mixture of the components with the mass ratio T:B = 10:5 (T—mica particles, B—water) was placed into an electrolyzer, in which a titanium plate served as a cathode and a graphite anode. The distance between the electrodes was 13 mm, the voltage across the electrodes was 220 V, the maximum current density was 0.2 A/cm^2^, and the duration of the experiment was 24 h. Then, wet grinding of the mica was carried out in the mill with a propeller stirrer equipped with cutting blades in the presence of a polymer bead filler. The number of the mixer revolutions was 800–900 rpm for 1.5–2 h [21,22]. The crushed material was subjected to classification with a separation of the working flakes of fraction 40–60 μm [23]. 

The source of the titanium (IV) required to coat the flakes was titanium sulfate salt—TiOSO_4_·H_2_O—which was obtained from the titanite mineral concentrate according to the well-known method [24,25]. The process was carried out in an electrodialysis cell with an MA-40 anion-exchange membrane (Shchekinoazot, Tula, Russia) (Figure 5).

Distilled water (50 mL) was poured into the cell cathode space, and 1 g of flakes was added while stirring. A 0.1 N sulfuric acid solution (V = 25 mm) served as anolyte. A direct current U = 10–15 V, I = 0.01 A was connected to the electrodes (cathode—titanium, anode—graphite) and the solution of titanium (IV) sulfate was gradually dosed into the catholyte through a separating funnel, the amount of which was taken at the rate of 15–20% TiO_2_ to the mass of the flakes. The end of the electrohydrolysis process was achieved when the catholyte pH reached 6–6.5. The resulting suspension was filtered and the precipitate was washed with distilled water. The precipitate was then heat-treated in the following two stages: first, it was dried at a temperature of 100 °C to remove any free water, and then it was calcined at 700–900 °C. The calcined product was a finely dispersed powder, silky to the touch, with a characteristic pearlescent sheen.

The phase composition of the samples were determined with the Shimadzu XRD-6000 X-ray diffractometer (Shimadzu corporation, Kyoto, Japan) using the ICDD-2019 database.

## 4. Results

Figure 6 shows the behavior of the main structure-forming mica cations in a media with a pH 3–9. It was noted that the leaching of the elements in the acidic pH range occurred more intensively, which is typical for trioctahedral micas [26].

The leaching of interlayer potassium from mica is controlled by an ion exchange between the potassium and a proton [27]; therefore, these reactions are accelerated in the acidic pH range. In the transition to the alkaline region, the leaching of the potassium ions gradually decreases and essentially does not change after reaching a value pH 8. For aluminum and silicon cations, the minimum degree of leaching occurs in the neutral region, which is associated with their amphoteric properties.

The X-ray diffraction patterns of the phlogopite samples prepared at different pH values are shown in Figure 6b. The XRD diffractograms of all the samples consist of phlogopite peaks. Based on the data shown in Figure 6a,b, we can assume that the migration of potassium from phlogopite in the acid medium does not only occur from the outer surface of the mica. As in an acidic environment, the peaks are weaker and broader. As we approach to the neutral medium, the intensity of the peaks increases and they become thinner. However, all these changes are insignificant and no significant distortion of the phlogopite crystal lattice is observed.

The initial rate of leaching of the structure-forming elements, from phlogopite into the solution, was rather high (Figure 7). The presence of surface defects in the particles treated at pH 3–5 was evidenced by a slight increase in the Ssp index by 1.5–2.0 m^2^/g; whereas in an alkaline medium, this indicator was practically non-existent, with changes in comparison with the initial value (9.06 m^2^/g).

Studies have shown that the raw material selected by the authors is sufficiently resistant to a prolonged exposure to the environment, and is therefore suitable for obtaining shell compositions. 

Traditional methods for the cleavage of mica are carried out by thermal, chemical and, much less frequently, electrochemical actions. External influences lead to a violation of the electroneutrality of the particles. In this case, the interlayer bonds are weakened and the surface activity increases [28]. Of the listed methods, the least effective is the thermal method, which is due to the removal of only crystallization water from the interlayer space [21]. Chemical cleavage leads to leaching of both the interplanar cations, in particular K, and the structure-forming ones, which causes the appearance of defects on the particle surface [29].

We used the method based on the electrochemical treatment of mica, since it excludes long-term contact of the particles with a chemical reagent (more than 100 h), and thereby prevents the formation of surface defects. Judging from the data in Figure 8, the degree of potassium extraction from phlogopite reached 13.5% with respect to K_2_O, within a time interval of 3 h. This figure is approximately 1.5 times higher than that necessary for breaking the electroneutrality of the packet, leading to its stratification [30].

For further experiments, a fraction of flakes of 40–60 µm, with a thickness of up to 1 µm, was used, which was obtained by grinding and a subsequent classification of the material [21]. Below are the SEM images of the mica particles. Figure 9 shows the SEM images of phlogopite particles, according to the processing scheme shown in Figure 4, as follows: 1—initial particles, a thick multi-layer flake with an irregular surface and sharp edges; 2—phlogopite particles after electrochemical splitting, thin flakes, with a smooth, even surface and smoothed edges; 3—particles after grinding, represented by different fractions (large and small); 4—particles after sieving, represented by the fraction of particles of 40–60 µm, suitable for further coating with titanium dioxide.

The mechanism of the process occurring in the chambers of the cell can be summarized as follows:At the cathode: 4H_2_O + 4e → 2H_2_ + 4OHAt the anode: 2H_2_O + 4e → O_2_ + 4H^+^In the cell: TiOSO_4_ +2OH^−^→TiO(OH)_2_ + SO_4_^−2^;4H^+^ +2SO_4_^−2^→ 2H_2_SO_4_TiOSO_4_ + 4H_2_O →TiO(OH)_2_ + H_2_SO_4_ + 2H_2_↑ + O_2_↑

In the experiments (Table 2), the amount of TiO_2_ introduced into the catholyte in relation to the mass of the mica particles was 18.5%. The phlogopite flakes in the catholyte served as crystallization centers for the solid phase of titanium (IV) hydroxide, which formed as a result of electrohydrolysis. However, depending on the process conditions, deposition of hydroxide can take place selectively on the flakes (pp. 1.2 Figure 10b), or it can additionally form an individual phase (pp. 3.4 Figure 10a), which leads to an undesirable effect (matting of the product) that degrades the quality of the PP.

Acceptable results were obtained at a current density on the electrodes of 0.01–0.015 A/cm^2^, and a process duration of 3–4 h. Under these conditions, the degree deposition of titanium (IV) on the mica flakes was 77–86%.

Further heat treatment of the electrohydrolysis product at 700–750 °C provided a product with a titanium dioxide coating of an anatase structure; when the temperature rose to 800–950 °C, a structural rearrangement occurred and rutile was formed. Figure 11 shows that the intensity of rutile increases and the peaks of rutile sharpen with the increasing calcination temperature, while the intensity of anatase decreases. When the calcination temperature is increased to 800 °C, the TiO_2_ in the pigments is in rutile form, so anatase is transformed completely to rutile. The area of use of the casing product depends on the structural variety of the coating.

An image of a phlogopite flake with a shell coating of titanium dioxide, obtained using a Philips XL 30 scanning electron microscope, is shown in Figure 12. The flake thickness—0.85 µm, and the coating thickness—200–250 nm.

## 5. Conclusions

The results obtained showed the possibility of using technogenic waste from the processing of mineral layered materials, in particular phlogopite, to obtain highly efficient functional products of a core–shell structure. The compositions with an inert core (micro flake of a mica) coated with mesoporous titanium dioxide of an anatase structure are of principle interest. According to the literature, this type of coating provide the processes of photocatalysis and surface self-cleaning. The compositions are more economically attractive than pure titanium dioxide; in particular, TiO_2_ grade P25, manufactured by Degusse.

The electrodialysis method for the coating of mica flakes has a number of advantages over the traditional method based on the thermohydrolysis of a titanium (IV) sulfate solution. The implementation of the original energy-saving method of coating in an electrodialysis cell simultaneously solves the environmental issues related to the disposal of acidic effluents, and is perfectly combined with the electrochemical method of mica splitting.

## Figures and Tables

**Figure 1 materials-14-03369-f001:**
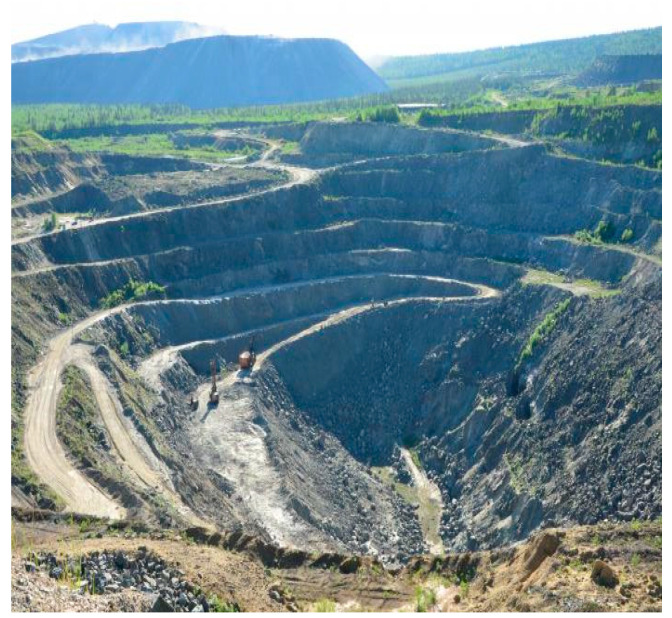
Enterprise quarry.

**Figure 2 materials-14-03369-f002:**
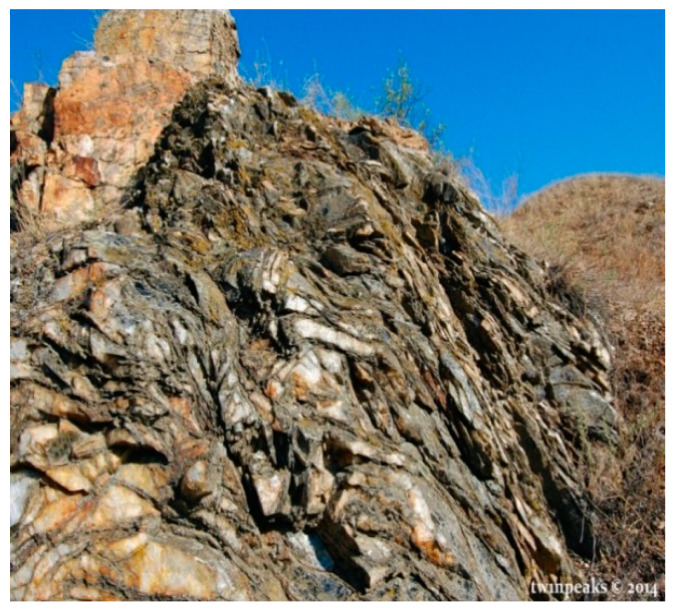
Manifestations of phlogopite clusters.

**Figure 3 materials-14-03369-f003:**
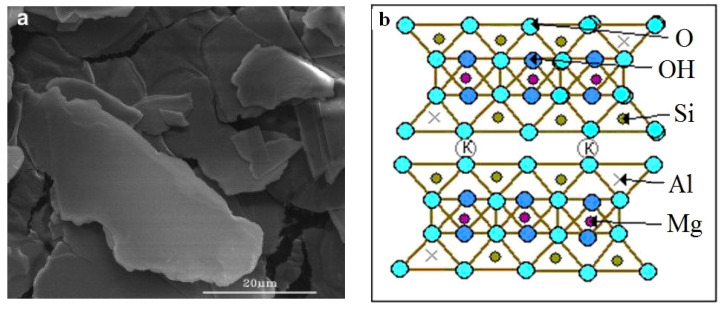
Phlogopite: (**a**) SEM images of mica particles; (**b**) crystal structure of mineral.

**Figure 4 materials-14-03369-f004:**
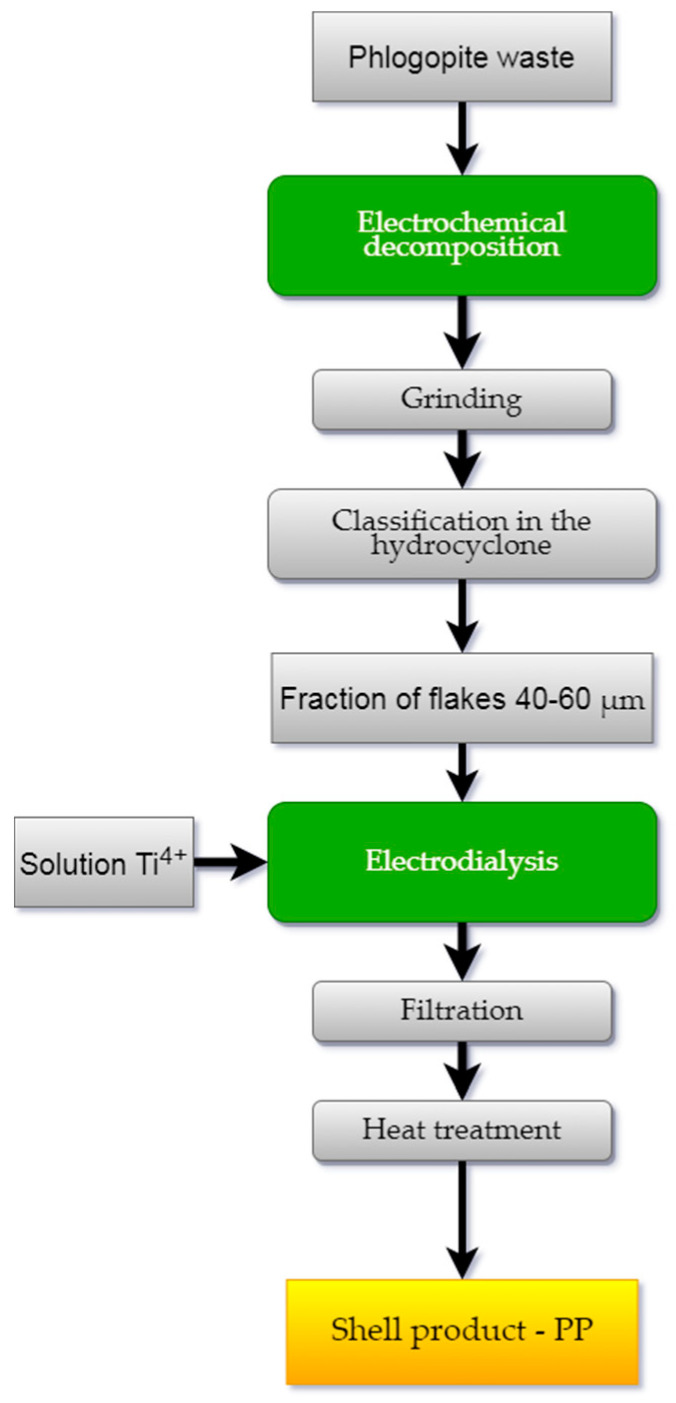
Electrochemical scheme of phlogopite waste processing.

**Figure 5 materials-14-03369-f005:**
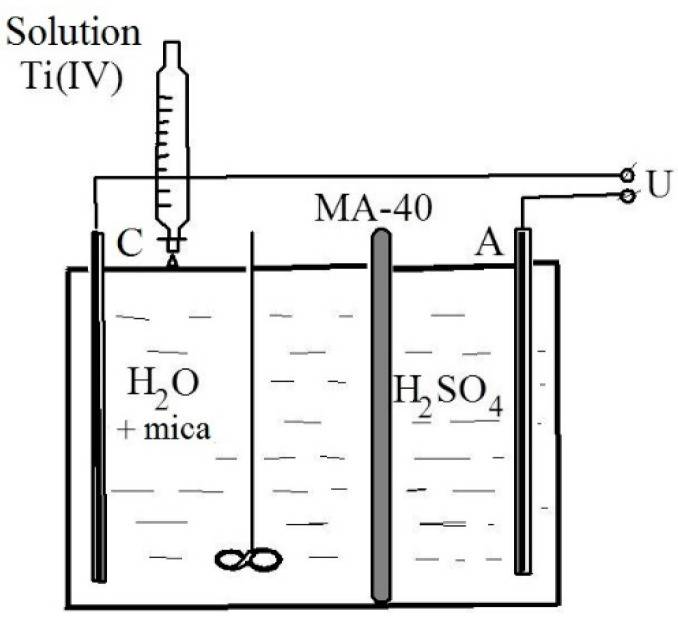
Electrodialysis installation.

**Figure 6 materials-14-03369-f006:**
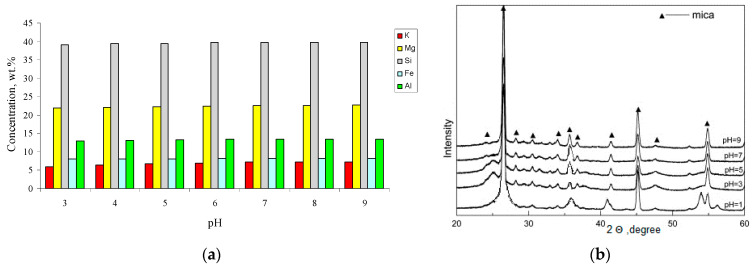
Concentration of main component of phlogopite with different initial pH values (**a**); X-ray diffraction patterns of mica with various pH values of 1, 3, 5, 7, and 9, respectively (**b**).

**Figure 7 materials-14-03369-f007:**
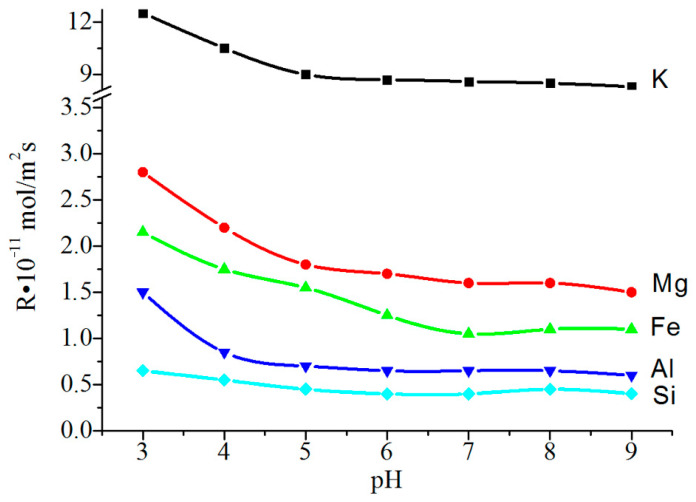
The initial rate of leaching of cations from phlogopite in the range of pH 3–9.

**Figure 8 materials-14-03369-f008:**
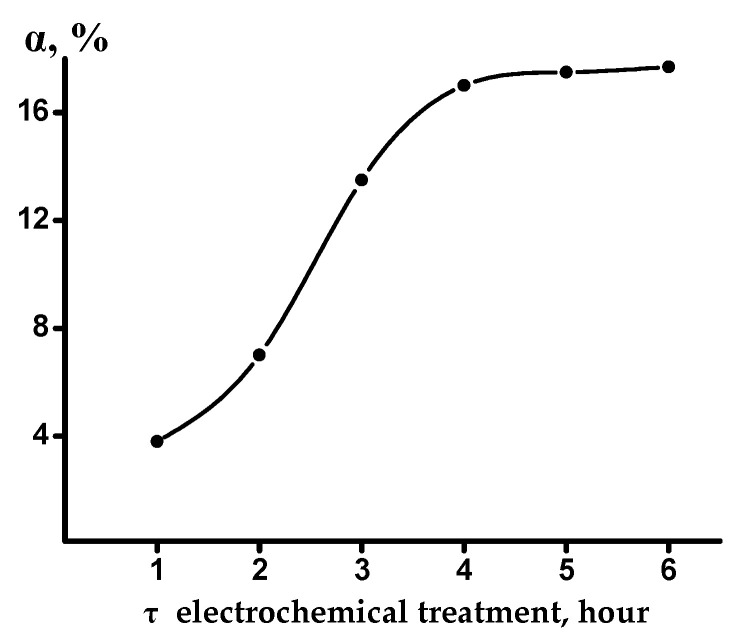
Potassium extraction (α, % K_2_O) from phlogopite by electrochemical treatment.

**Figure 9 materials-14-03369-f009:**
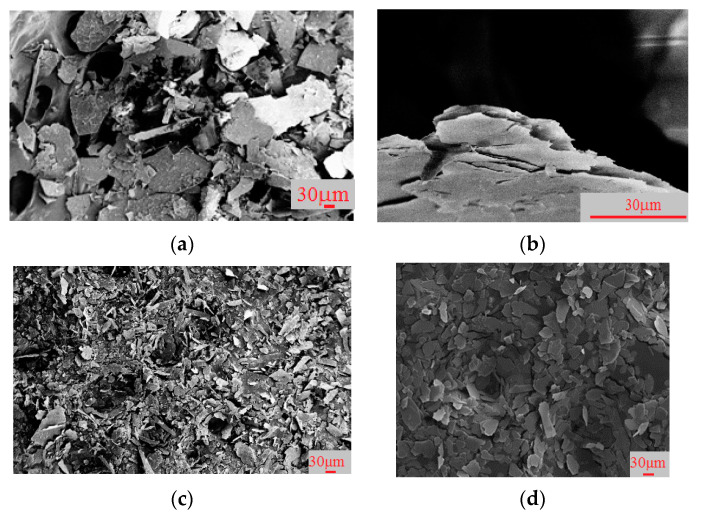
SEM images of mica particles. (**a**) Initial; (**b**) particles after electrochemical treatment; (**c**) particles after grinding; (**d**) after sieving. Translated with www.DeepL.com/Translator (free version).

**Figure 10 materials-14-03369-f010:**
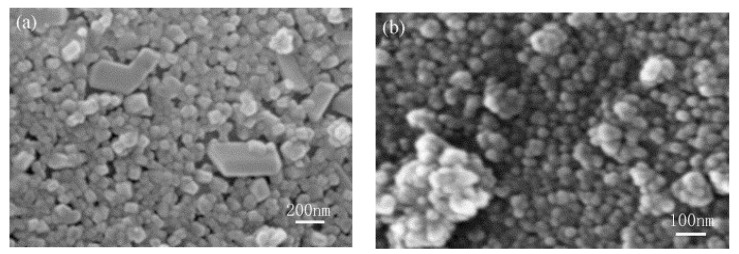
SEM micrographs of the morphology of the TiO_2_ thin layers on mica particles (**a**) sample obtained in experiments 3, 4; (**b**) sample obtained in experiments 1, 2.

**Figure 11 materials-14-03369-f011:**
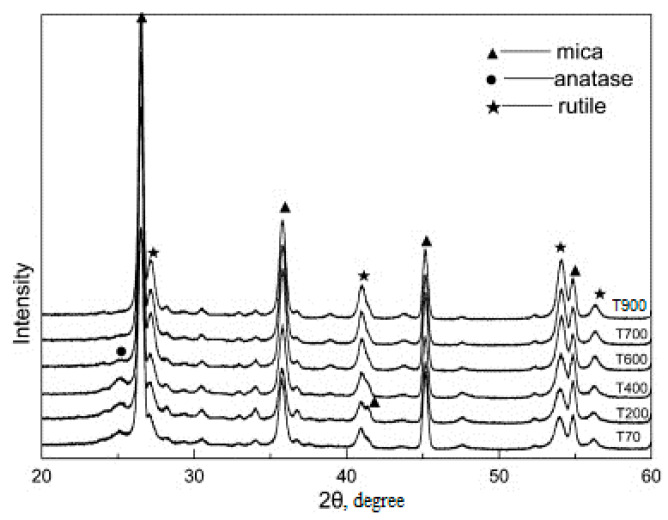
X-ray diffraction patterns obtained for the pigments calcined at the following different temperatures: 70, 200, 400, 600, 700, 900 °C.

**Figure 12 materials-14-03369-f012:**
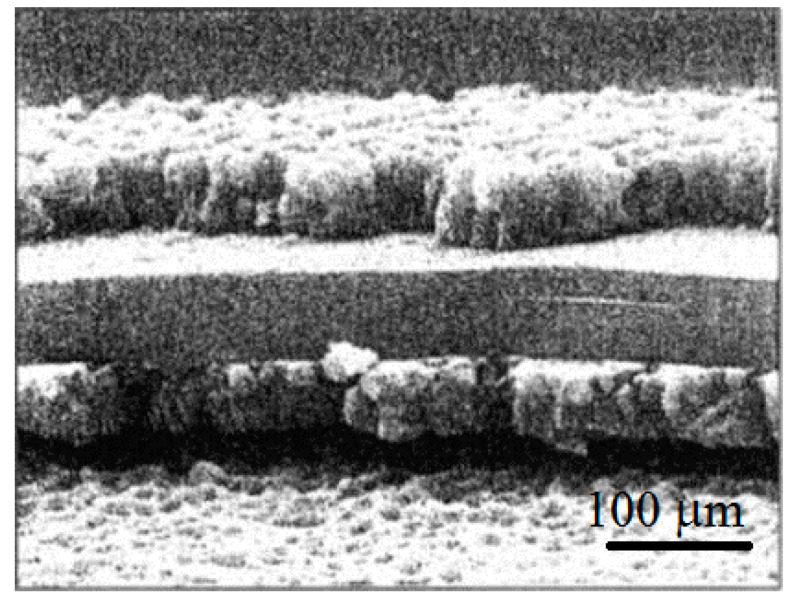
SEM image of the phlogopite flake coating of titanium dioxide.

**Table 1 materials-14-03369-t001:** Chemical composition of phlogopite.

Main Components	Contents, wt%
SiO_2_	40.0 ± 0.1
Al_2_O_3_	13.5 ± 0.1
MgO	23.2 ± 0.1
Fe_2_O_3_	8.35 ± 0.01
K_2_O	9.53 ± 0.01
CaO	0.158 ± 0.001
Na_2_O	0.439 ± 0.001
Specific surface area, m^2^/g	9.06

**Table 2 materials-14-03369-t002:** Conditions of electrodialysis and the degree of deposition of TiO_2_ on the flakes.

No.	Current Density, A/cm^2^	Time, h	Amount of TiO_2_ on the Surface of the Flakes, wt. %	Amount of TiO_2_ in the Form of Hydroxide, wt. %	Degree of Deposition on the Flakes, %
1	0.010	4.5	14.3	3.1	77.3
2	0.015	3.0	15.9	2.0	85.9
3	0.020	2.2	6.0	12.1	31.7
4	0.050	1.0	4.9	12.5	26.5

## Data Availability

The data presented in this study are available on request from the corresponding author.

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
