# Peer review of "Mineral Layer Fillers for the Production of Functional Materials"

_materials, 2021, doi:10.3390/ma14123369_

Round 1

Reviewer 1 Report

Please find the attachement. 

Author Response

Thanks to the reviewer for carefully reading the article. Our responses to reviewer comments.

Point 1. It would be beneficial for readers to include information about crystal structure of mica — space group, crystal system, lattice parameters, symmetry

Response 1. Phlogopite is the most magnesian mineral of the biotite group. The crystals are tabular, short prismatic in shape. The crystal structure is layered lattice. Class of symmetry is prismatic - 2/m. Syngony is monoclinic. Spot group is 2/m – Prismatic. The spatial group is B2/m (B1 1 2/m) [C2/m] {C1 2/m 1}. Muscovite is a rock-forming mineral from the mica subclass of laminated silicates. Muscovite splits easily into thin sheets, which is due to its crystalline structure, composed of 3-layer packages of 2 sheets of silica and alumina tetrahedrons connected through a layer composed of octahedrons, with Al ions in the centre, surrounded by 4 oxygen ions and 2 OH groups; 1/3 of the octahedrons is not filled with Al ions. The bundles are interconnected with potassium ions. Spot group is 2/m – Prismatic. The spatial group is B2/b (B1 1 2/b) [C2/c] {C1 2/c 1}. Syngony is monoclinic.

Point 2. The Authors wrote: The initial material, consisting of lamellar particles of 10 – 12 mm, was ground in a porcelain ball mill using the wet mode. A fraction of 63 – 100 microns was used for the work. How did the Authors measure those 63 – 100 microns and why it is important?

Response 2: The particle size of the mica after grinding was determined at the hydrocyclone. Particles larger than 100 microns were regrind. There are standard particle size requirements for pearlescent pigments, the finer the particles the better. We have chosen a medium particle size 63-100 microns.

Point 3. In Table 1 the Authors show the chemical composition of phlogopite. In my opinion, besides the chemical analysis, it is essential to check the structural properties of the materials on each stage of the experiment. Such measurements would support chemical composition investigations and give a better overview on the topic since the properties of mica and mica-TiO2 are related to their crystal structure.

Response 3: X-ray diffraction patterns showing structural changes have been added to the text of the article.

Point 4. The Authors wrote: A mixture of the components with the mass ratio T:B = 10:5. . .

It needs to be defined what are T and B in that ratio.

Response 4: At the mass ratio T:B = 10:5, T - the mica particles, B – water.

Point 5. In Figure 6 the Authors show the concentration of the elements in [mg/g] as a function of pH. It would be useful for readers to recalculate those values and show them also in [%].

Response 5: We changed the Figure 6.

Point 6. The data in Figure 7 should be described in the figure (Si, Al, . . . ). Please consider showing them also in [%] as mentioned before.

Response 6: We made some changes to the figure7. However, it is not possible to go to the mass percentages because we are talking about the leaching rate of the mica components into the solution and we are analysing the solution.

Point 7. The vertical axis in Figure 7 is labeled in Russian. Please, describe it in English.

Response 7: The vertical axis in Figure 7 is translated.

Point 8. The Author wrote: (. . . ) contact of the particles with a chemical reagent (more than 100 h), and thereby prevents the formation of surface defects. Why it prevents the formation of surface defects? How the Author can tell that? It has to be explained.

Response 8.  Surface defects form during prolonged contact with chemical reagents, leaching of structure-forming components takes place. «Holes» and potholes form, the mica flakes become cloudy, the titanium dioxide is irregularly applied to the surface. The functional properties of the pearlescent pigments are impaired

Point 9. Please, label the vertical axis in Figure 8.

Response 9. The Y axis is titled

Point 10. In Figure 9 the scale bars are in Russian. Please, label the pictures in English. Also, the images have different scales. For the sake of clarity and comparison, please, show the data with the same scale bar.

Response 10. Figure 9 has been modified.

Point 11. The Authors wrote: Further heat treatment of the electrohydrolysis product at 700 – 750°C provided a product with a titanium dioxide coating of an anatase structure; when the temperature rose to 800 – 950°C, a structural rearrangement occurred and rutile was formed.

How can the Authors tell that the anatase or rutile phases were formed? In my opinion, it needs to be proved by other methods, such as x-ray diffraction studies (or other techniques). This remark is connected to the previous one where I pointed out that the structural analysis should be carried out on each stage of the experiment to prove that the material under investigation has desirable structural properties.

Response 11. X-ray diffraction patterns showing structural changes have been added to the text of the article.

Point 12. Out of 30 references, 12 are in Russian. It is a huge inconvenience for readers who do not speak Russian. In my opinion, for better understanding and to provide a broader overview on the topic, it is essential to include references written in English.

Response 12.  5 references added to the reference list

Reviewer 2 Report

Dear Authors,

Thank you for your submission, please follow my comments given in .pdf file.

I suggest mainly to improve the overall quality of presentation and discussion (+conslusions), clarify the information of TiO2 phases  and add some more SEM micrographs of powders produced with various process parameters. The other comments are in .pdf file.

Author Response

Thanks to the reviewer for carefully reading the article. Our responses to reviewer comments.

Reviewer 3 Report

Submitted manuscript is focused on new method of using technogenic wasts for photocatalytic application. The method is well desribed. I see the only weakness in language.

Author Response

Thanks to the reviewer for carefully reading the article

The text of the article has been corrected

Round 2

Reviewer 1 Report

  •   The Authors included the XRD patterns in the manuscript. However, there is no explanation or discussion in the text. For example --- does pH changes lattice parameters of mica or anatase/rutile? Are there any (micro)structural differences related to the chemical process? What is the origin the reflections not indexed in the Figures? In my opinion, it should be described in the text to provide better scientific background to the work done in the paper. Additionally, there is no information about the diffractometer and its configuration used in the experiments.
  • The Authors included Fig. 6a, as they were asked.  However, similar to the previous case, there is no discussion of the results. Are the changes in chemical composition important? Do they have any influence on the properties of the material? It seems that there is no significant influence, however, it should be mentioned and discussed somewhere in the text. 

Author Response

Thanks to the reviewer for carefully reading the article. The text has been revised

Reviewer 2 Report

Dear Authors,

I am relatively fine with the revisions provided. However, some improvement in a few places are needed:

  • Fig. 6, XRD diffractograms were added but there is no discussion to Fig. 6b. This should be added, especially if some differences between diffractograms may be observed (like for example for theta of ~25 or 55 etc.).
  • The same for Fig. 9 (a-d), a discussion needed. The most informative is the figure label "1 - initial; 2 - particles after electrochemical treatment; 3 - particles after grinding; 4 – after sieving" but this is not enough.
  • Place a completely new scale bars on ALL SEM micrographs, please. They should be readable and in the same format of "μm".

Author Response

Thanks to the reviewer for carefully reading the article. The text of the article has been corrected.

Responce 2: for the sake of clarity, the authors cannot put the image on the same scale
